

# Validation of torus mapping method for dealiasing Doppler weather radar velocities

Peter Smerkol[1], Vito Švagelj[1], Anton Zgonc[1], and Benedikt Strajnar[1,2]

[1]Slovenian Environment Agency, Vojkova 1b, SI-1000 Ljubljana, Slovenia
[2]University of Nova Gorica, Vipavska 13, SI-5000 Nova Gorica, Slovenia

**Correspondence:** Peter Smerkol (peter.smerkol@gov.si)

**Abstract.** In this paper, the implementation and validation of the torus mapping, a method to dealias radial wind observations by meteorological radars is presented. We apply the procedure to data from the Operational program for exchange of weather radar information (OPERA) over central part of Europe for the whole year 2021. The quality of the resulting radar winds is assessed by a comparison to colocated radiosonde and aircraft observations and also through analysis of first guess departures

5 using a regional numerical weather prediction model. Performance of quality control on dealiased radial wind data is also evaluated. We show that the torus mapping method is a robust procedure on data with a sufficiently low amount of measurement noise. It produces datasets of comparable quality for a wide range of Nyquist velocity values and has potential for operational applications.

10 **1 Introduction**

Radial velocity of scatterers can be measured with a Doppler radar via the time-dependent phase shift of the scattered electromagnetic wave. Because the phase difference is a circular variable, defined on the interval $[-\pi, \pi]$, there exists a maximal value of radial velocity that can be measured unambiguously, called the Nyquist velocity. Since the phase shift is measured between consequent radar pulses, the Nyquist velocity can be expressed as:

$$15 \quad v_{ny} = \frac{\lambda}{4\tau_p} = \frac{\text{PRF} \cdot \lambda}{4}, \tag{1}$$

where $\tau_p$ is the time between pulses, PRF is the pulse repetition frequency and $\lambda$ is the radar wavelength.

Radial velocities larger in magnitude than the Nyquist velocity will be folded (aliased) back into the interval $[-v_{ny}, v_{ny}]$. Depending on the velocity value, multiple aliasing can occur, so the true (dealiased) velocity $v_r$ can be expressed as:

$$v_r = v_o + 2nv_{ny}, \tag{2}$$



where $v_o$ is the observed radial velocity and $n$ is an unknown integer called the Nyquist number, which indicates the number of foldings that occurred.

Dealiasing the data therefore comes down to determining the correct Nyquist number for each observation. To achieve this, multiple postprocessing methods with different approaches have been devised. Aliasing causes abrupt changes in the radial velocity field where the Nyquist number changes. These changes occur only in rare atmospheric situations, so methods are mostly based on the assumption that the radial velocity is a smooth quantity and derive the Nyquist number using this assumption.

Early approaches were based either on checking the continuity with single radial or 2D statistics, with one or more passes through the data (Ray and Ziegler (1977); Bargen and Brown (1980); Miller et al. (1986); Liang et al. (1997); Eilts and Smith (1990)), or on variational techniques, where regions with the same Nyquist numbers are identified (Merritt (1984); Bergen and Albers (1988); Jing and Wiener (1993); Wüest et al. (2000)). These methods need additional external wind data, e.g. from a nearby measured wind profile or from a numerical weather prediction (NWP) model.

Based on (Eilts and Smith, 1990) and (Bergen and Albers, 1988), modern Next Generation Weather Radar (NEXRAD) algorithm and China New Generation Weather Radar (CINRAD) with its CINRAD dealiasing algorithm (CINDA) (Zhang and Wang (2006); He et al. (2012a); He et al. (2012b)) methods were developed. These methods are not dependent on additional external data.

An alternative method, also not dependent on additional wind information, has been developed (Haase and Landelius, 2004), based on (Siggia and Holmes, 1991). The method solves the aliasing problem by mapping measurements onto a surface of a torus. The same method was used in this paper and is briefly described in the next section.

Aliasing limits the usefulness of radial wind measurements which are often performed in reflectivity-optimised way, where range is preferred over the quality of wind measurements. This is the case for numerous data providers of the Operational Program for Exchange of weather RAdar Information (OPERA, Saltikoff et al. (2019)), a project run by the European Meteorological Services Network (EUMETNET). Our goal is to investigate if radar measurements, after being dealiased by the torus mapping method, can be of sufficient quality. A comparison between dealiased radar winds and measurements from aircraft and radiosondes is made. Next, the three measurement types are compared with results of a meteorological model, used as an independent reference, to explicitly show that the dealiased radar measurements can be used in data assimilation for NWP and in other applications.

## 2 Torus mapping method

The torus mapping method is described following (Haase and Landelius, 2004). Like other methods, this method assumes a smooth radial wind velocity field, specifically a linear field in zonal ($v$) and meridional ($u$) wind speeds for a specific height. We can express this wind model as:

$$v_r \approx v_m(u,v) = (u\sin\alpha + v\cos\alpha)\cos\phi, \tag{3}$$





where $\alpha$ is the azimuth angle and $\phi$ is the elevation angle.

At a given elevation angle and distance from the radar, the wind model radial velocity as a function of azimuth is a sinusoidal curve with a form that depends on the zonal and meridional wind components. Because of aliasing, the observed radial velocity curve can have discontinuities (Fig. 1 left).

55

Since both the observed radial velocity and azimuth are circular coordinates, the first on the interval $[-v_{ny}, v_{ny}]$ and the second on the interval $[0, 2\pi]$, we can make the domain of the curve doubly periodic on these intervals and map the curve onto this domain, or in other words, map the radial velocity measurements onto the surface of a torus (Fig. 1 right).

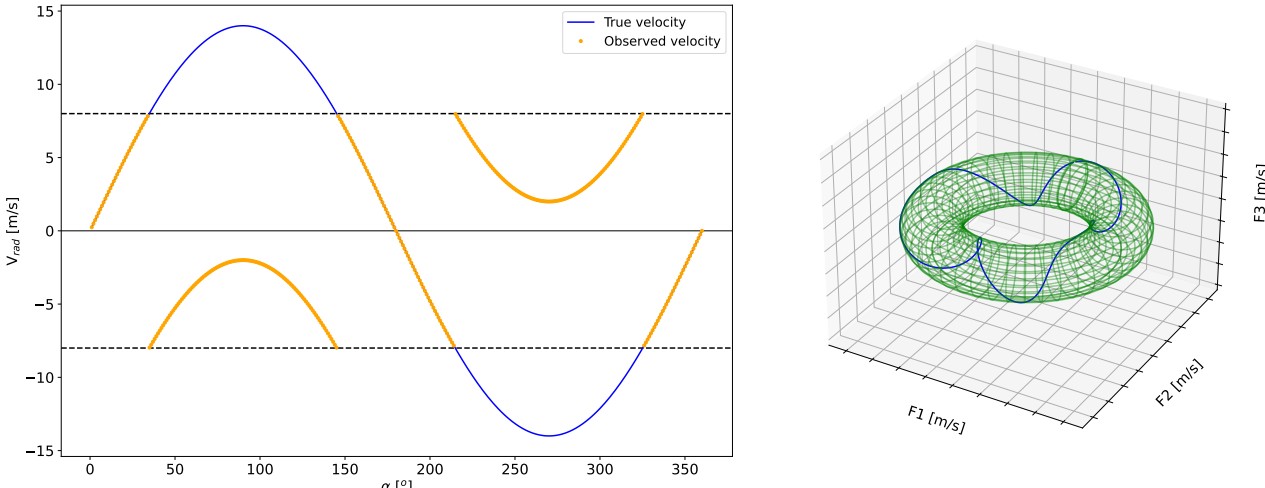

**Figure 1.** Idealized wind field observations from a Doppler weather radar with Nyquist velocity of $8\,\mathrm{ms}^{-1}$. Left: Azimuth dependence, both observed and true velocities are shown. Right: Same observations mapped onto a torus surface. It can be seen that there are no discontinuities on the right plot.

The mapped curve can be expressed as a continuous parametric curve:

60
$$\boldsymbol{F}(\alpha) = \left( \left[ R + \frac{v_{ny}}{\pi} \sin\left( v_o \frac{\pi}{v_{ny}} \right) \right] \sin\alpha, \left[ R + \frac{v_{ny}}{\pi} \sin\left( v_o \frac{\pi}{v_{ny}} \right) \right] \cos\alpha, \frac{v_{ny}}{\pi} \cos\left( v_o \frac{\pi}{v_{ny}} \right) \right), \tag{4}$$

where $R$ is the torus radius, which has to satisfy $R > v_{ny}/\pi$, but is otherwise arbitrary.

Using Eq. (2), it can be seen that the aliased ($v_o$) and dealiased ($v_r$) velocities map to the same points on the curve. Using this fact and the wind model from Eq. (3), one can find an expression for the azimuth derivative for the third component of the curve in Eq. (4):





$$D \quad = \frac{\partial F_3}{\partial \alpha} = -au + bv, \tag{5}$$

$$a \quad = \cos\alpha \cos\phi \sin\left(v_o \frac{\pi}{v_{ny}}\right), \tag{6}$$

$$b \quad = \sin\alpha \cos\phi \sin\left(v_o \frac{\pi}{v_{ny}}\right). \tag{7}$$

Coefficients $a$ and $b$ can be calculated for all data points and derivative $D$ can be estimated from data with any numerical method that estimates derivatives. Furthermore, the method can be used separately for any subset of the whole dataset, which allows us to divide the dataset into subsets that better satisfy the linear wind condition from Eq. (3). For each subset, components $u$ and $v$ can be independently determined by a least squares approach:

$$\{u, v\} = \min_{u,v} \sum_{k=1}^{N} \left[D_k - (-ua_k + vb_k)\right]^2, \tag{8}$$

where $N$ is the number of points in the subset, and the index $k$ goes through every point in the subset.

For every point $k$ in the subset, one can then find the correct Nyquist number by minimizing:

$$n = \min_{n} |v_{o,k} + 2nv_{ny,k} - v_{m,k}|, n \in \mathbb{Z}. \tag{9}$$

## 3 Validation methods and datasets

### 3.1 Algorithm implementation effects

The torus mapping method was implemented in the Python 3 programming language. Since the implementation details can influence the results, they are described in detail below.

The derivatives $D_k$ in Eq. (8) are calculated with central differences:

$$D_k = \frac{\partial F_3(\alpha_k)}{\partial \alpha_k} = \frac{F_3(\alpha_{k+1}) - F_3(\alpha_{k-1})}{\alpha_{k+1} - \alpha_{k-1}}. \tag{10}$$

As a consequence, all measurements that do not have two neighbouring measurements in the azimuth direction are left out. These mostly represent measurement noise and edges of precipitation (see Fig. 3).

Noise exclusion generally improves the torus mapping method, as noise predominantly contains random values, centered around zero, which causes large fluctuations in numerical derivatives. Also it does not follow the linear wind assumption from Eq. (3). This causes the minimization in Eq. (8) to converge to the wrong values, which was verified using an artificial dataset (Fig. 2), where it can be seen that after introducing noise to the dataset, the minimization converges to the wrong values and the value error increases.

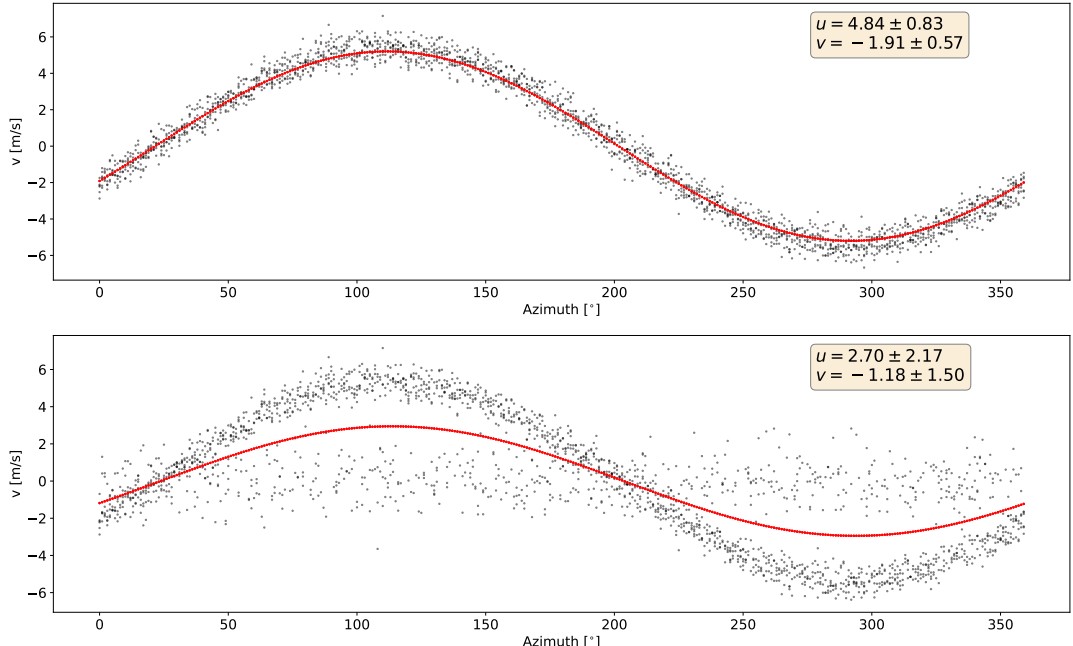

**Figure 2.** The minimization procedure on an artificial dataset with 2500 points. Upper plot shows data with $u = 5\,\mathrm{ms}^{-1}$, $v = -2\,\mathrm{ms}^{-1}$ with added Gaussian noise with spread $0.5\,\mathrm{ms}^{-1}$ and lower plot has 30 % of points replaced with a Gaussian noise centered around 0 with spread $1.0\,\mathrm{ms}^{-1}$.

The algorithm divides data into subsets within $100\,\mathrm{m}$ height intervals. Any height interval that contains less than 500 measurements, or has the maximum determined value of the wind model $v_m$ bigger than $60\,\mathrm{ms}^{-1}$, is rejected, to ensure a better percent of dealiased data for which minimization in Eq. (8) converged (see Fig. 3).

## 3.2 The OPERA radar dataset

The European radar network consists of national datasets with unique observation practices. This makes the combined data set largely inhomogeneous in terms of age, frequency band, manufacturer, software, scanning strategy and calibration, all limiting the possibility to use the entire dataset without a specific treatment per radar network or even individual site. OPERA (Saltikoff et al., 2019) addresses these issues by providing homogenized measurements in a standard data format. The Opera Data Centre (ODC) has been providing volume data suitable for data assimilation in NWP since 2012, for over 200 radar sites of 25 European countries, including numerous sites that also measure Doppler winds on top of the radar reflectivity. In this study we validate the entire dataset from the year 2021, as operationally downloaded and archived at the Slovenian Environment Agency



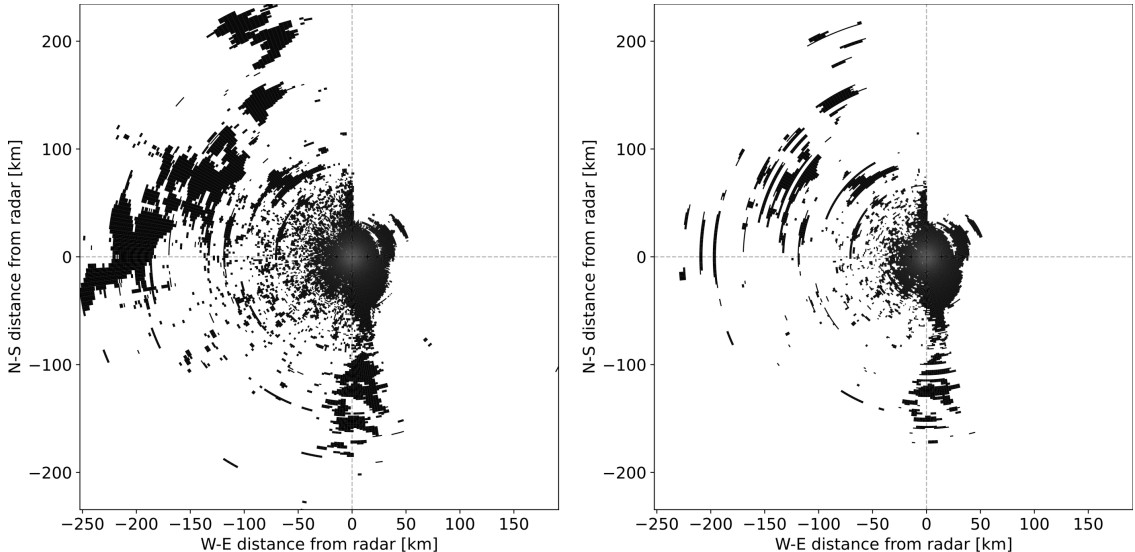

**Figure 3.** Locations of aliased (left) and dealiased (right) radial wind measurements relative to radar for one radar scan. Exclusion of noise, precipitation edges and data in specific height intervals due to the algorithm implementation can be seen.

(ARSO). Some providers do not include information about the Nyquist velocity or do not include radial wind measurements at all. After excluding these providers, we are left with wind data from 8 radar networks, which also differ by Nyquist velocity value, from $7.3\,\mathrm{ms}^{-1}$ to $64.1\,\mathrm{ms}^{-1}$ (see Fig. 4).

### 3.3  In situ upper-air wind observations

Several in-situ upper-air wind measurements are available to validate the dealiased Doppler winds, including the radiosondes,
accurate but relatively sparse observations, and in particular the aircraft observations. The latter have seen major advances over Europe in the last few years, mainly through successful exploitation of the so-called mode selective (Mode-S, de Haan (2011)) derived data, on top of the well-established Automatic Meteorological DAta Relay (AMDAR, Painting (2003)). Several flavours of Mode-S observations are being used at ARSO and in this study: the major European Mode-S EnHanced Surveillance (Mode-S EHS, de Haan (2011)) dataset, processed and provided by the European Meteorological Aircraft Derived Data Center
(EMADDC), and additional Mode-S Meteorological Routine Air Report (Strajnar, 2012) datasets originating from the air-traffic control authorities in Slovenia and Czech Republic. The density of aircraft data is much higher than that of radiosondes, making it much more suitable for validation of radar winds. All these inputs are subject to quality control and/or whitelisting at the level of data providers before entering the NWP-based control within the data assimilation procedure.

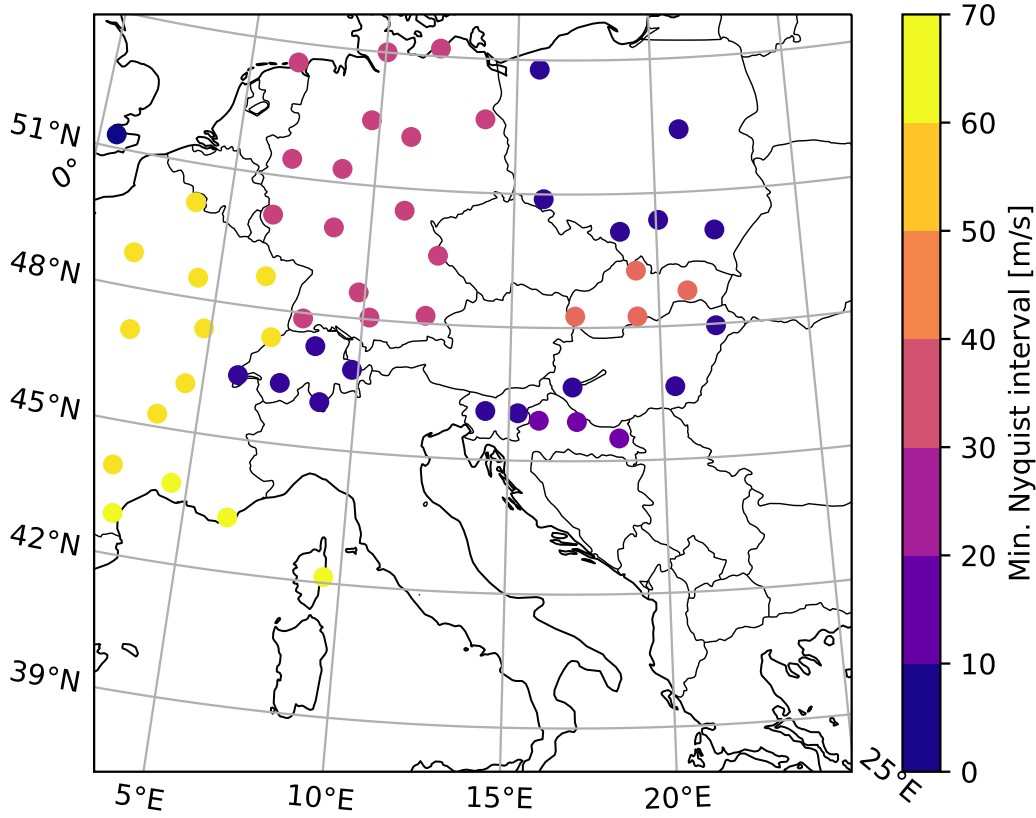

**Figure 4.** The operational ALARO/SI model domain and OPERA radar sites used in the study. Colors represent the minimal Nyquist interval of the provided radial winds.

### 3.4 Numerical weather prediction model

To further evaluate the quality and usability of dealiased Doppler winds, the data assimilation component of NWP model at ARSO was also used. In its current main operational version, the ALARO (ALadin-AROme, Termonia et al. (2018)) canonical system configuration of the ACCORD (A Consortium for COnvection-scale modeling Research and Development) is used with a horizontal resolution of 4.4 km and 3 h data assimilation cycling. Figure 4 shows the operational ALARO model domain and the radar sites within this domain. The archived operational 3 h first guesses were used as inputs to the observation screening

process, the initial step of the three-dimensional variational assimilation (3D-Var, Fischer et al. (2005)), within which the first-guess (or background) departures in observation space are calculated for each individual measurement, including Doppler



winds, aircraft and radiosondes. The departures and the accompanying quality control flags were computed sequentially for all analysis times in 2021 and stored in the observational database for further analysis and mutual collocations.

### 3.5 Colocation and validation techniques

Fully overlapping upper-air atmospheric measurements are rare. For conventional observation platforms, some spatial and temporal distance must be ensured for the sake of their safe operations. Doppler wind measurements, on the other hand, are mostly constrained to the areas where precipitating hydrometeors are present; they are also found in the areas of fog and mist, typically in morning and evening inversion layers, and sometimes in so-called angel echoes. Aircraft sometimes try to avoid these areas when precipitation intensity is high. For an intercomparison between observations, a certain spatial and

temporal distance must be allowed in order to gain adequate sample sizes. On the other hand, this separation introduces the natural atmospheric variability in the difference statistics. Although a triple-collocation approach (de Haan, 2016) would be preferable, we choose here the pair-wise collocation, mainly because the radiosondes are very sparse compared to the other two data sets. The allowed horizontal separation in this study is $10\,\mathrm{km}$ and the vertical difference can be up to $100\,\mathrm{m}$, which is in agreement with other similar studies (de Haan (2011); Strajnar (2012); Mirza et al. (2016)). The temporal interval is $\pm 5$

minutes around the NWP analysis time. The aircraft and radiosonde measurements and the Doppler wind on the other hand differ on the number of wind components they provide. In order to enable valid intercomparison, the radial component was computed separately for aircraft and radiosondes, with respect to the radar sites from which the collocated radar was measured.

The comparison against the NWP model, on the other hand, enables a comparison in terms of the observed variable, with the model counterpart simulated by the observation operators from the first guess. Here the model serves as an independent

(yet imperfect) reference for assessment of the relative quality of individual observations, without affecting their quality since the final analysis is not performed and the model is not cycled in time.

Furthermore, we can use the quality control procedure as part of data assimilation in ALARO model to improve the quality of comparison. Since it is designed for identification of erroneous observations, it can reject either aliased or wrongly dealiased data. This is also an important consideration for possible operational implementation of dealiased radial winds. Currently, the

rejection threshold for radar data in ALARO is set to $20\,\mathrm{ms^{-1}}$ difference with respect to model background, which makes it more efficient for data sets with higher Nyquist velocities.

## 4 Results

### 4.1 Choice of radar datasets for validation

The whole dataset was divided into two parts according to the magnitude of Nyquist velocity. The first part contains radar obser-

vations from countries with smaller minimum Nyquist velocities ($\lesssim 12\,\mathrm{ms^{-1}}$), where we expect a big influence of dealiasing. To be sure that only influences from dealiasing is studied, only observations from two Slovenian radars are retained, because they contain the least amount of noise which influences the outcome of the dealiasing procedure (see Fig. 2). We then also have





only one small minimum value of Nyquist velocity in the dataset ($8\,\mathrm{ms}^{-1}$), which makes the dealiasing effects clearer. This part of the dataset we call Dataset A. The second part contains observations from German, French and Slovakian radars, which

have larger Nyquist velocities ($\gtrsim 30\,\mathrm{ms}^{-1}$) and a small influence of the dealiasing procedure is expected, but is nevertheless useful to study. This part we call Dataset B.

### 4.2    Comparison of colocated observations

A comparison of differences in colocated observations of radial wind for Dataset A and Dataset B is shown in Fig. 5, where for radiosondes and aircraft the radial wind was calculated from measured $u$ and $v$ wind components with respect to the

colocated radar site. Distributions of differences for radar-radiosonde (RSD) and radar-aircraft (RAD) pairs are plotted, with the distribution of aircraft-radiosonde (ASD) differences added for reference. ASD contains wind magnitudes instead of radial winds, since there is no colocated radar observation to serve as a reference for radial wind computation. All distributions are normalized by number of colocated observations for each category for easier objective comparison. Relevant statistical parameters about the distributions are gathered in Table 1. In all plotted distributions, there is an expected central peak centered

around $0$ and a spread that reflects the joint measurement errors of observation types used in each difference distribution and natural variability due to observation displacements. In aliased distribution of Dataset A, there are two clearly visible side peaks with centers at two times the Nyquist velocity ($-16$ and $16\,\mathrm{ms}^{-1}$) in RAD, and less visible in RSD, which is expected because it contains less events. These two peaks indicate that we have a non-negligible amount of aliased data with Nyquist numbers $-1$ and $1$ in Dataset A, which also reflects in the spread of data, which is $3.54$ times wider for RAD and $3.29$ times

wider for RSD than in ASD. After dealiasing, data from side peaks in Dataset A is shifted to the main peak, and the result is a single peak which is $1.28$ times wider for RAD and $1.56$ times wider for RSD than the reference ASD. In aliased data of Dataset B, side peaks with aliased data still occur, but are significantly smaller in comparison to the central peak than in Dataset B (only visible in the logarithmic scale), which is expected, since data with larger Nyquist velocities contain less aliased data. The peaks are centered around $-65$ and $65\,\mathrm{ms}^{-1}$, which is twice the value of Nyquist velocity of the German radars. Spread

of aliased data is $1.96$ times wider for RAD and $2.24$ times wider for RSD compared to ASD. After dealiasing, these peaks are partially shifted to the central peak, thus reducing the spread to $1.13$ times wider for RAD and $1.64$ times wider for RSD compared to ASD. The dealiasing algorithm also reduces the number of radar observations for about one fifth in both datasets.

### 4.3    Comparison of first guess departures

In the second step, the colocated observation pairs were evaluated against the ALARO NWP model. This was performed

by running the initial phase of data assimilation procedure, called screening, to compute the first guess departures, i.e. the differences between observations and their model counterparts, as calculated by observation operators. Comparison of first guess departure distributions (FGDD) was done only for radar and aircraft observations, because radiosonde data was too sparse. Here, we compare departures as provided by the model, for individual wind components $u$ and $v$ measured by aircraft and for radial wind measured by radar, taken as a single wind component. As before, apart from the central peak of nonaliased

data, present in all distributions, radar FGDD for aliased dataset A contains two side peaks with aliased data corresponding to





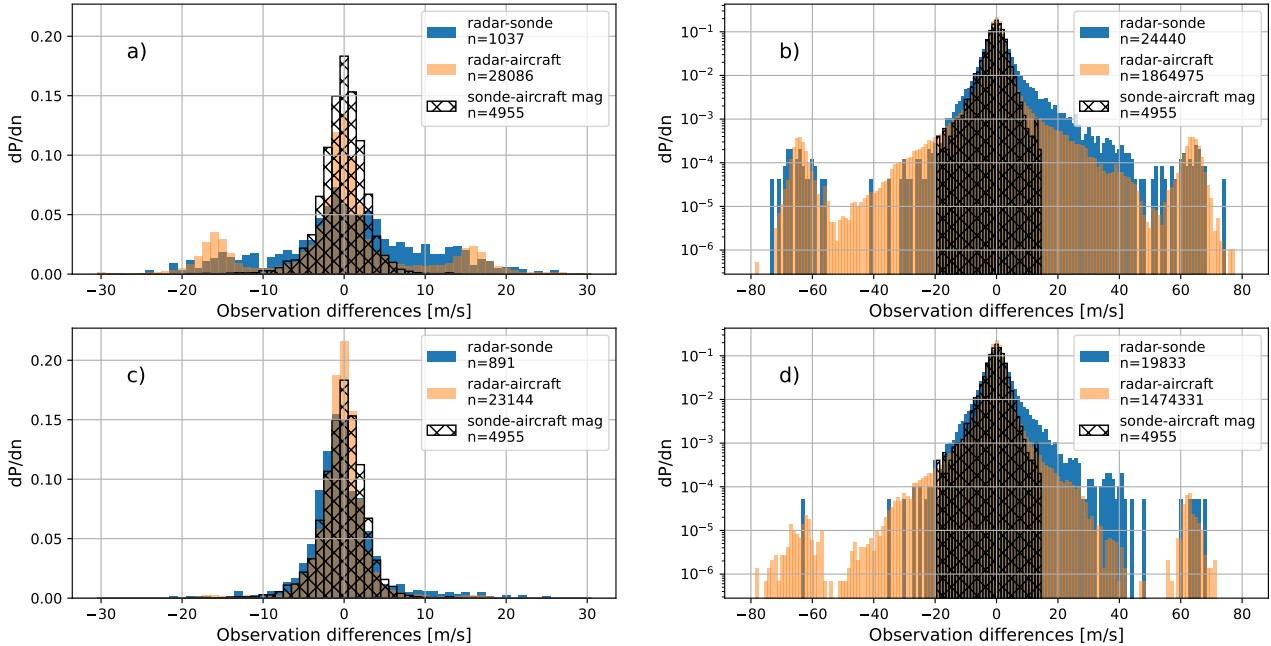

**Figure 5.** Differences between colocated pairs of observations (ASD, RAD, RSD) for (a) aliased Dataset A, (b) aliased Dataset B, (c) dealiased Dataset A and (d) dealiased Dataset B. Note the linear scale on y axis for Dataset A and logarithmic scale for Dataset B.

| | Aliased data | | | Dealiased data | | |
|---|---|---|---|---|---|---|
| Case | N | avg [ms$^{-1}$] | std [ms$^{-1}$] | N | avg [ms$^{-1}$] | std [ms$^{-1}$] |
| ASD A | 4955 | -0.19 | 2.89 | 4955 | -0.19 | 2.89 |
| RAD A | 28086 | -0.76 | 10.23 | 23144 | -0.26 | 3.71 |
| RSD A | 1037 | 0.73 | 9.51 | 891 | -0.11 | 4.52 |
| ASD B | 4955 | -0.19 | 2.89 | 4955 | -0.19 | 2.89 |
| RAD B | 1864975 | 0.06 | 5.66 | 1474331 | -0.06 | 3.28 |
| RSD B | 24440 | 0.69 | 6.48 | 19833 | 0.27 | 4.73 |

**Table 1.** Relevant statistical parameters for distributions analyzed in comparison of colocated observations.

Nyquist numbers $-1$ and $1$ (see Fig. 6). In the dealiased radar FGDD for Dataset A, these side peaks are almost completely shifted to the central peak, thus reducing the ratio of standard deviations from radar and (average) aircraft FGDD from 3.01 to 1.22. All relevant statistics about the plotted FGDD are shown in Table 2. For Dataset B, the amount of aliased data in the sample is expectedly smaller, but dealiasing still reduces the ratio defined above from 2.09 to 1.25.




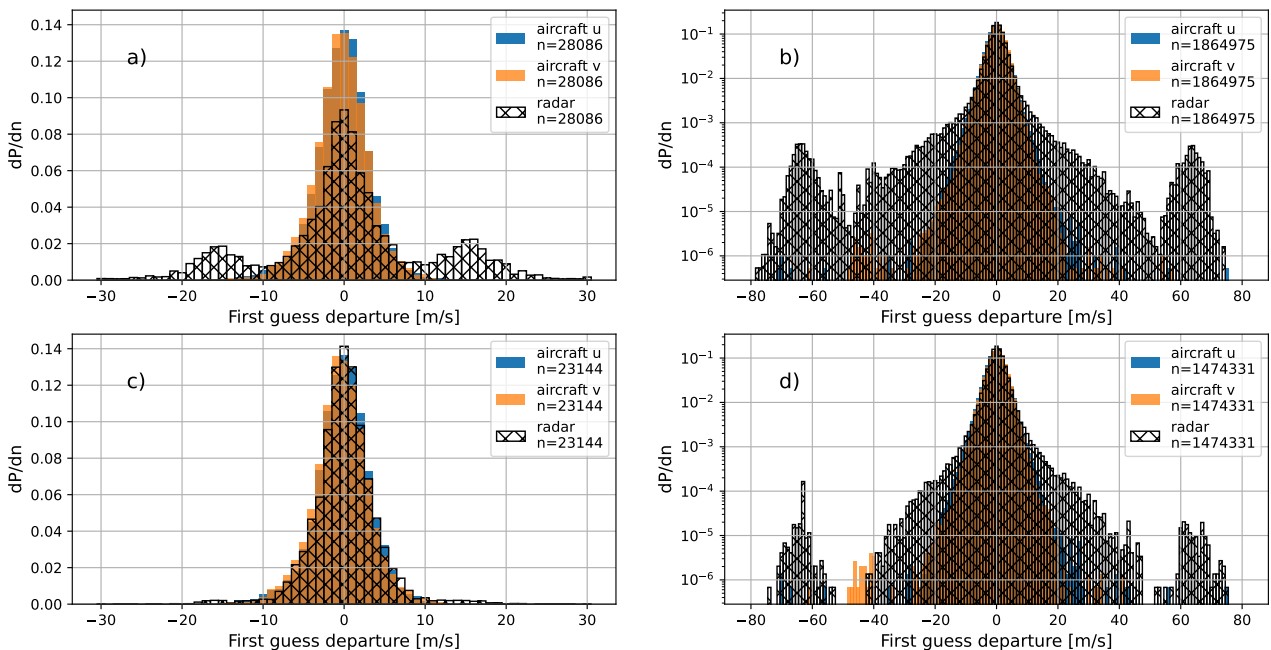

**Figure 6.** Aircraft and radar FGDD for colocated radar and aircraft pairs for (a) aliased Dataset A, (b) aliased Dataset B, (c) dealiased Dataset A and (d) dealiased Dataset B. Note the linear scale on y axis for Dataset A and logarithmic scale for Dataset B.

| | Aliased data | | | Dealiased data | | |
|---|---|---|---|---|---|---|
| Case | N | avg [ms$^{-1}$] | std [ms$^{-1}$] | N | avg [ms$^{-1}$] | std [ms$^{-1}$] |
| aircraft u FGDD A | 28086 | -0.19 | 3.44 | 23144 | -0.17 | 3.4 |
| aircraft v FGDD A | 28086 | -0.33 | 3.43 | 23144 | -0.35 | 3.45 |
| radar FGDD A | 28086 | 0.66 | 10.33 | 23144 | 0.12 | 4.18 |
| aircraft u FGDD B | 1864975 | -0.01 | 2.78 | 1474331 | -0.02 | 2.83 |
| aircraft v FGDD B | 1864975 | 0.12 | 2.7 | 1474331 | 0.12 | 2.74 |
| radar FGDD B | 1864975 | 0.02 | 5.72 | 1474331 | 0.14 | 3.49 |

**Table 2.** Relevant statistical parameters for distributions analyzed in comparison of first guess departures for colocated pairs.

In the next step, we compare the aliased and dealiased radar FGDD for a dataset containing observations from Slovenian radars (SI dataset) and a dataset consisting of observations from German radars (DE dataset), where instead of colocated pairs, all available radar observations are taken into account. This is useful since the data is not preselected and contains measurements from all possible situations including all high wind events. This can highlight effects of multiple aliasing for datasets with a




smaller Nyquist velocity or the value of dealiasing procedure on datasets where aliasing is less frequent because of a larger
value of Nyquist velocity $\gtrsim 30\,\mathrm{ms}^{-1}$.

In Fig. 7a, comparison for SI dataset is shown. Slovenian radars have two values of Nyquist velocity, one at around $8\,\mathrm{ms}^{-1}$
for lower radar elevations and the other at around $40\,\mathrm{ms}^{-1}$, used in higher radar elevations. In both FGDD, we see multiple
side peaks, corresponding to Nyquist numbers from $-4$ to $4$ for the smaller $v_{ny}$ value and Nyquist numbers $-1$, $1$ for the
larger $v_{ny}$ value. Side peaks in dealiased FGDD are decreased by about one order of magnitude, except the peaks at Nyquist
numbers $-4$ and $4$, which are increased, but because of the large aliasing of data in these peaks, this effect likely comes from
wrong dealiasing. The dealiasing algorithm significantly narrows the FGDD central peak by shifting aliased data into it, which
reduces the standard deviation of FGDD from $9.7\,\mathrm{ms}^{-1}$ to $4.7\,\mathrm{ms}^{-1}$ (see Table 3). The algorithm rejects about $24$ percent of
data.

In DE dataset in Fig. 7b has only one value of Nyquist velocity at around $32\,\mathrm{ms}^{-1}$, which results in a clear side peak in
FGDD at $64\,\mathrm{ms}^{-1}$, corresponding to Nyquist numbers $-1$ and $1$. As in the SI dataset, side peaks are decreased by an order
of 10 by shifting data from them to the central peak. Since DE dataset contains less aliased data, the central peak is narrower,
but not significantly. However, the standard deviation of FGDD by dealiasing is reduced from $5.39\,\mathrm{ms}^{-1}$ to $3.29\,\mathrm{ms}^{-1}$. In this
case, the dealiasing algorithm reduces the dataset by 19 percent.

| Case | Aliased data | | | Dealiased data | | |
|---|---|---|---|---|---|---|
| | N | avg [ms$^{-1}$] | std [ms$^{-1}$] | N | avg [ms$^{-1}$] | std [ms$^{-1}$] |
| SI dataset | 5885707 | -0.44 | 9.7 | 4490567 | 0.12 | 4.7 |
| DE dataset | 40455234 | 0.09 | 5.39 | 32705072 | 0.1 | 3.29 |

**Table 3.** Relevant statistical parameters for distributions analyzed in comparison of first guess departures for all observations.

## 4.4 Interaction of dealiasing and quality control in data assimilation

Figure 8 shows the radar FGDD for previously selected colocated pairs of radar and aircraft observations, split into raw and
active observations after the first guess quality check. For aliased Dataset A, it can be clearly seen that a large part of aliased
data is retained and would enter active assimilation in the current ALARO setup. It can be seen that the shifting of aliased
data into the central peak by the dealiasing procedure has two effects. Firstly, the previously accepted aliased observations
remain in the active data set with corrected value, and secondly, the previously rejected data from the distribution tails are no
longer flagged as outliers but become active. In Dataset B, acceptance of aliased data is not an issue, since Nyquist velocities
are bigger, but an improvement is still made due to the second effect mentioned above, thus increasing the percentage of
accepted observations. Most notably, the peaks at around $\pm 65\,\mathrm{ms}^{-1}$ are reduced by a factor of 10. The acceptance rate (ratio
of active and raw observations) for Dataset A improves significantly due to dealiasing, from $94.7\%$ to $99.8\%$. For Dataset B
the improvement is smaller (from $98.9\%$ to $99.7\%$), as is to be expected.





**Figure 7.** Radar FGDD for all observations for (a) aliased SI dataset, (b) aliased DE dataset, (c) dealiased SI dataset and (d) dealiased DE dataset. Note the logarithmic scale on y axis for both datasets.



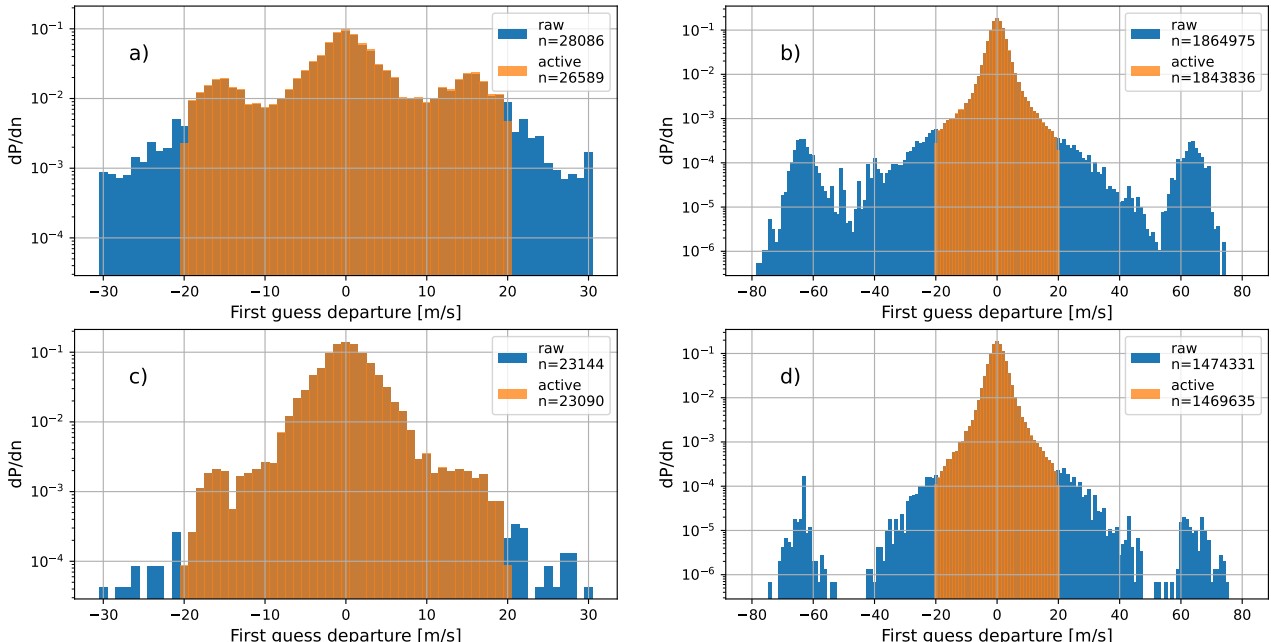

**Figure 8.** Radar FGDD for colocated radar and aircraft pairs for raw and active observations after the first guess quality check for (a) aliased Dataset A, (b) aliased Dataset B, (c) dealiased Dataset A and (d) dealiased Dataset B. Note the logarithmic scale on y axis for both datasets.

## 5 Discussion and conclusions

Weather radars can measure reflectivity and radial wind velocity which are exclusive measurements in the sense that for reflectivity measurements, a small PRF is desired to reach the maximum range, whereas for radial velocity measurements, a small PRF implies a small Nyquist velocity, which increases the amount of aliasing. This can be solved by implementing separate wind optimized scans, however, this is often not a viable scanning strategy. The other solution is to implement a reliable dealiasing algorithm. This paper presents the first systematic evaluation of the torus mapping algorithm (Haase and Landelius, 2004) on a large dataset over Europe over a time period of one year. The main goal is to show the usefulness of dealiased radial winds by comparing them to well established upper-air wind observations used for data assimilation in operational NWP and the NWP model itself.

Our validation shows that the dealiasing procedure significantly improves the quality of radar wind measurements. Since the aliased side peaks in distributions of differences, both with respect to other established wind observations and the NWP model, are reduced by an order of magnitude, we can roughly estimate that the procedure correctly dealiases about 90% of aliased data. To evaluate this improvement quantitatively, we compare standard deviations of distributions of colocated pairwise differences of radial wind, aircraft and radiosonde measurements. Such distributions include the natural variability of the wind





field originating from horizontal and vertical displacements between measurements, on top of individual measurement errors.

In this way, we can use the aircraft-sonde differences as a reference, since these measurements are already known for good quality and are assimilated operationally. We show that for more aliased data (Dataset A), the standard deviation of differences involving radar measurements reduces from $10.23\,\mathrm{ms^{-1}}$ to $3.71\,\mathrm{ms^{-1}}$ and from $9.51\,\mathrm{ms^{-1}}$ to $4.52\,\mathrm{ms^{-1}}$ for aircraft and radiosonde data, respectively. This makes the standard deviations of distributions involving radar measurements comparable to the aircraft-sonde distribution ($2.89\,\mathrm{ms^{-1}}$), from which we conclude that the error of dealiased radar wind data is of the same

order as that of aircraft or radiosondes. Surprisingly, even for data, where aliasing is rare (Dataset B), improvement is still substantial, the standard deviations for aircraft and radiosonde reduce from $5.66\,\mathrm{ms^{-1}}$ to $3.28\,\mathrm{ms^{-1}}$ and from $6.48\,\mathrm{ms^{-1}}$ to $4.73\,\mathrm{ms^{-1}}$, respectively, again making these statistics much more comparable to aircraft-radiosonde reference. A comparison of the same colocated datasets against the NWP model as a reference was also performed. Here, the difference distributions include the model forecast error instead of natural variability, which allows for a similar but independent intercomparison

of difference distributions. It also allows for a separate evaluation for each observation type. As seen from a comparison of Table 1 and Table 2, very similar conclusions can be also drawn for this verification method. These results suggest that the dealiasing procedure increases the quality of radar wind observations to a level comparable to aircraft and radiosonde data. Such a procedure is mandatory for radars with low Nyquist velocity ($\sim 10\,\mathrm{ms^{-1}}$) and furthemore results show that even for radars having a large Nyquist velocity ($\gtrsim 30\,\mathrm{ms^{-1}}$), the dealiasing of data can be beneficial. This is additionally confirmed on

a very large dataset from the German radar network, where first guess departures for all data (not just colocated pairs) were analyzed.

As shown, the correct convergence of the torus mapping algorithm is very dependent on the amount of noise in the data. While the algorithm itself removes some noise by using central differences for calculation of derivatives, we suggest that the very noisy datasets are not suitable for dealiasing in this way. Better results are also achieved on data comprised of large

connected precipitating regions instead of sparsely dispersed ones and data with good coverage in the azimuth direction. Algorithm is also less successful in weather situations where the linear wind assumption is not fully satisfied, such as vicinity of fronts, near the center of cyclones or strong convective precipitation. Our analysis shows that for this reasons, data is wrongly dealiased or not dealiased in around $10\%$ of the situations (see Fig. 7).

Ideally, these erroneous observations would be rejected by quality control performed as a part of data assimilation in NWP.

We show that in the default setting of ALARO quality control, the first side peaks in aliased data of Dataset A (small Nyquist velocity) are accepted, therefore, even after dealiasing, a part of wrongly dealiased data gets assimilated. To mitigate this, we propose a stricter quality control threshold for radial wind data assimilation in this case. For Dataset B (higher Nyquist velocity), quality control will reject all remaining data from aliased side peaks, but assimilation still benefits from dealiasing because of higher acceptance rate.

In the current work, we show that dealiasing with the torus mapping method is a robust procedure that produces datasets of sufficient quality, comparable to other already established tropospheric wind observations, and has potential for operational applications. It significantly improves data measured with low Nyquist velocities but is applicable on any data with some degree of aliasing, provided that the amount of measurement noise is sufficiently low. While it would be optimal to use



dedicated wind scans, these are not available everywhere neither foreseen in the near future and also have a drawback of
having shorter maximum range. So by using dealiasing, which may become part of centralized processing at OPERA, much
more measurements of the European radar network can become available for use in data assimilation. A dedicated study is
planned in order to evaluate the impact of this additional radar data on the regional NWP forecast.

*Code and data availability.* Implementation of torus mapping algorithm and the code for artificial wind model dataset creation, together
with two sample OPERA radar HDF5 files from Dataset A and Dataset B are available at https://doi.org/10.5281/zenodo.7816818 (Smerkol
et al., 2023). Datasets used for the validation (OPERA radar data sets for whole year 2021, aircraft and radiosonde observations, short-range
ALARO model forecasts) are archived and available upon request from the Slovenian Environment Agency. Terms and conditions apply for
the following data sets: access to OPERA volume radar data is restricted to member institutions of EUMETNET, and access to Mode-S EHS
aircraft observations is subject to a non-disclosure agreement with the European Meteorological Aircraft Derived Data Center (EMADDC,
https://emaddc.com/default.aspx).

*Author contributions.* PS and VŠ wrote the software, PS, BS, VŠ and AZ developed the methodology, PS, BS, VŠ performed the validation
and analysis, VŠ made the visualisations, PS wrote the original draft preparation and BS, VŠ and AZ reviewed the draft.

*Competing interests.* The authors declare that they have no conflict of interest.



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
