# Peer review of "Validation of torus mapping method for dealiasing Doppler weather radar velocities"

_EGUsphere, 2023_

## Referee Comment (RC1)

**Review: Validation of torus mapping method for dealiasing Doppler weather radar velocities**

**Summary:**

This manuscript introduces a dealiasing method based on torus mapping and evaluates this with a larger dataset. Torus-mapping requires assumptions about the linearity and homogeneity of the wind field that limit the potential applications of this method.

**General comments:**

1. The main application is a bit unclear. Is the target data assimilation? There is little to no discussion of particularly challenging dealiasing situations, such as high-shear situations, or when data is sparse. A plethora of algorithms handle large-coverage linear wind situations very well, so the added benefit here is rather unclear.
2. What are advantages over existing methods? There are very few references to current dealiasing methods, especially also for the countries that the data stems from (e.g. DE, CH, FR). How do the assumptions and limitations of torus-mapping compare to VAD-based methods such as IVAP (Liang et al., 2019)?
3. The specifications of the used data are unclear. Why introduce the data of CH, GB, PL, HR, HUN if they are not used? Fig. 4 misleadingly implies that all of this data was used for evaluation. How are the dual- and triple PRF data from DE and FR dealt with? Do they deliver the multi-PRF extended Nyquist range product? These are known to have problems with noise and errors that require different corrections than Nyquist-interval-based dealiasing. If dataset A only contains Slovenian radars, how is it different from dataset SI – apart from the filtering for events?
4. The paper would benefit from having at least one explicit data example for the low Nyquist velocity datasets. E.g. instead of just showing the areas in Fig. 3, the actual velocity values would be very helpful.
5. Given the potential operational implementation, what are the computational requirements and estimated runtime of the algorithm for a full volume?

**Specific comments:**

1. Line 3: In **the** central part of Europe
2. Lines 27-35: This fails to include more recent advances in dealiasing (see reference recommendations below). In addition, some of these methods rely on the VAD, which is also derived from the radar itself and hence not an external data source.
3. Line 48: More recent methods apply this criterion only very locally. How does your approach fair in high-shear situations? How does it depend on having large data coverage?
4. Section 2: How does this compare to the VAD method (Germann, 1999 and Tabary et al., 2001) and VAD-based dealiasing, such as the IVAP method (Liang et al., 2019)?
5. Lines 84ff: What does this imply for the use of this method? What is your target application and how does it suffer from this limitation? E.g. it appears that this method would not work very well in high turbulence / high shear situations, as e.g. found in mountainous countries or convective situations.
6. Line 89: So do you divide each elevation into areas that are within 100m altitude slices or do you add data from other elevations?

7. Section 3.2: Here you introduce the complete OPERA dataset, but you only use a fraction for the actual validation. This is misleading.
   Given that most countries that deliver Doppler data already have operational dealiasing, there should be a discussion of the novel method vs. the existing ones. Do countries also deliver dealiased data or only raw aliased velocities? Evidently some processing is applied to the dual- and triple PRF data of DE and FR before data delivery.
8. Line 151: Given that you exclude the majority of low-Nyquist radars, a more detailed explanation would be appreciated. Why do you introduce them in the first place?
9. Figure 5 and following: It would help a lot to include the labels Dataset A, Dataset B, aliased and dealiased in the figure itself. (I.e. over the columns and next to the rows).
10. Figure 5 and following: The hatching in the panels covering +-80 m/s makes it difficult to see the details. Could you remove the vertical lines in the histogram?
11. Fig. 6: Presumably you also have peaks at multiples of the Nyquist velocity in log-space?
12. Fig. 7: Please use the same scale for both panels here, there is no evident reason to change the scale here.
13. Line 226: The generally more interesting application is to low-Nyquist velocity datasets, where you only evaluate 2 radars – this is not a large European dataset. The evaluation of the high-Nyquist velocity radars is of course still beneficial, but the error rates are much lower in the first place.
14. Line 229: How does this improvement in acceptance rate compare to other methods? E.g. how high is the dealiasing failure rate?
15. Line 238: The evaluation is very indirect and based on external wind measurements that are fundamentally different from Doppler velocity measurements. Did you also check the number of apparent folds in e.g. the gate-to-gate-velocity difference?
16. Line 253: Given the noise in multi-PRF data, are they even suitable here? How about applying denoising techniques prior to dealiasing?
17. Line 256: **The** algorithm …
18. Line 256ff: If all of these situations are excluded, what is the main application for this method and why is it to be preferred over others that handle high-shear situations better? 10% failure rate seems extremely high, given that dealiasing failure rates are usually <1% (Louf et al., 2020 and Feldmann et al., 2020).

**Suggested references:**

- Liang, X., Y. Xie, J. Yin, Y. Luo, D. Yao, and F. Li, 2019: An IVAP-Based Dealiasing Method for Radar Velocity Data Quality Control. *J. Atmos. Oceanic Technol.*, **36**, 2069–2085, https://doi.org/10.1175/JTECH-D-18-0216.1.
- Hengstebeck, T., K. Wapler, D. Heizenreder, and P. Joe, 2018: Radar Network–Based Detection of Mesocyclones at the German Weather Service. *J. Atmos. Oceanic Technol.*, **35**, 299–321, https://doi.org/10.1175/JTECH-D-16-0230.1.
- Tabary, P., F. Guibert, L. Perier, and J. Parent-du-Chatelet, 2006: An Operational Triple-PRT Doppler Scheme for the French Radar Network. *J. Atmos. Oceanic Technol.*, **23**, 1645–1656, https://doi.org/10.1175/JTECH1923.1.
- Feldmann, M., C. N. James, M. Boscacci, D. Leuenberger, M. Gabella, U. Germann, D. Wolfensberger, and A. Berne, 2020: R2D2: A Region-Based Recursive Doppler Dealiasing

Algorithm for Operational Weather Radar. *J. Atmos. Oceanic Technol.*, **37**, 2341–2356, https://doi.org/10.1175/JTECH-D-20-0054.1.

- Louf, V., A. Protat, R. C. Jackson, S. M. Collis, and J. Helmus, 2020: UNRAVEL: A Robust Modular Velocity Dealiasing Technique for Doppler Radar. *J. Atmos. Oceanic Technol.*, **37**, 741–758, https://doi.org/10.1175/JTECH-D-19-0020.1.
- James, C. N., and R. A. Houze , 2001: A Real-Time Four-Dimensional Doppler Dealiasing Scheme. *J. Atmos. Oceanic Technol.*, **18**, 1674–1683, https://doi.org/10.1175/1520-0426(2001)018<1674:ARTFDD>2.0.CO;2.
- Germann, U., 1999. Vertical wind profile by Doppler radars. *MAP Newsletter*, *11*(2).
- Tabary, P., G. Scialom, and U. Germann, 2001: Real-Time Retrieval of the Wind from Aliased Velocities Measured by Doppler Radars. *J. Atmos. Oceanic Technol.*, **18**, 875–882, https://doi.org/10.1175/1520-0426(2001)018<0875:RTROTW>2.0.CO;2.

---

## Referee Comment (RC2)

Review of "Validation of torus mapping method for dealiasing Doppler weather radar velocities", by P. Smerkol et al

This paper presents a Doppler dealiasing technique based on torus mapping, which assumes that horizontal wind components are linear. The paper is well written and clearly organized. However, there are several flaws listed below that led me to reject this paper. The two main reasons for not recommending major revisions are that I don't believe such technique can ever produce error rates sufficiently low in the most important convective situations for operational applications, especially when low Nyquist velocities are used (because it's based on a linear assumption) and the underlying idea is the same as already-published work based on the VAD technique.

1. The torus method presented in this paper is identical to the idea of dealiasing Doppler velocity using VAD winds (Browning and Wexler 1968). Besides, the implementation using first-order derivatives of radial velocity is identical to the work presented in Tabary et al. (2001). These relevant papers are not cited anywhere in the manuscript.

2. The most challenging meteorological situations for dealiasing are associated with convective systems, especially in high shear environments and when small Nyquist velocities are used. Those are also the most important conditions operationally, where Doppler velocities need to be accurately dealiased. In these conditions, the horizontal wind components are far from linear, so the main assumption made in the torus / VAD analyses is not satisfied. This major problem is not even mentioned in the presentation of the method. Depending on the Nyquist velocity used, linear winds are expected to perform very poorly for dealiasing purposes (I have tested that myself). This is not evaluated at all in the paper using simulated winds with various degrees of non-linearity. Probably because the main finding will be that the technique cannot work in these convective situations.

3. The introduction is missing most major recent developments in dealiasing techniques (e.g., Helmus and Collis 2016; Feldmann et al. 2020; Louf et al. 2020) and some important older ones (James and Houze 2001).

4. Figure 3 shows that most of the precipitation data are not even considered for dealiasing, which does not make sense. A centred-difference scheme to estimate the first-order derivatives should not remove so many points, so I suspect that there is a bug in the implementation. Besides, a dealiasing technique is supposed to provide a solution for all measured Doppler velocity bins.

5. The validation with in-situ aircraft winds will only contain non-convective cases, since aircraft will avoid convective situations (especially at take-off and landing). Consequently, the validation shows that in non-convective situations, the technique works reasonably well. But all existing dealiasing techniques will work well in those simple situations.

6. When choosing datasets A and B, it seems like you are combining radar data in each set with different Nyquist velocities? If that is indeed true, it makes the interpretation of the PDFs of difference very difficult. Also, a much better metrics to estimate the accuracy of the technique than PDFs of difference would be to estimate error rates (i.e., how often is the technique wrong when Doppler data should or should not have been dealiased), see for instance Louf et al. (2020). Small percentages of wrongly dealiased points won't really show in mean and standard deviations of differences.

7. In the conclusion, the authors estimate that the procedure correctly dealiases about 90% of aliased data, which is a 10% error rate. This is definitely not good enough, especially operationally. The authors do not seem to realize that at all. Advanced techniques have error rates in the 0.1 to 0.2% (again, see recent publications).

8. The comparison with NWP based on rejected points with a 20 m/s threshold is interesting because it includes convection. However, the issue with using NWP is that it generally does not fully resolve convective-scale dynamics (even with a 4.4km grid) and convective systems are generally not at the right place. These issues are clearly discussed in the papers mentioned above, but not acknowledged anywhere in this paper.
9. I did not provide detailed editorial comments, given my recommendation to reject the paper.

References:

Browning, K. A., and R. Wexler, 1968: The determination of kinematic properties of a wind field using Doppler radar. J. Appl. Meteor., 7, 105–113.

Feldmann, M., C. N. James, M. Boscacci, D. Leuenberger, M. Gabella, U. Germann, D. Wolfensberger, and A. Berne, 2020. R2D2: A Region-Based Recursive Doppler Dealiasing Algorithm for Operational Weather Radar. J. Atmos. Oceanic Technol., 37, 2341-2356.

Helmus, J. J., and S. M. Collis, 2016: The Python ARM Radar Toolkit (Py-ART), a library for working with weather radar data in the Python Programming Language. J. Open Res. Soft., 4, e25, https://doi.org/10.5334/jors.119.

James, C. N., and R. A. J. Houze, 2001: A real-time four-dimensional Doppler dealiasing scheme. J. Atmos. Oceanic Technol., 18, 1674–1683.

Louf, V., A. Protat, R. A. Jackson, and S. M. Collis, 2020: UNRAVEL: a modular velocity dealiasing algorithm for C-band Doppler radar. J. Atmos. Oceanic Tech., 37, 741-758, DOI: 10.1175/JTECH-D-19-0020.1.

Tabary, P., G. Scialom, and U. Germann, 2001: Real-time retrieval of the wind from aliased velocities measured by Doppler radars. J. Atmos. Oceanic Technol., 18, 875–882.

*Alain Protat*

*Bureau of Meteorology, Melbourne, Australia*

*2 October 2023.*

---

## Author Comment (AC1)

**Response to RC1: Validation of torus mapping method for dealiasing Doppler weather radar velocities**

The authors would like to thank the Referee 1 for a very detailed review and constructive comments.

We agree with referee's remark that the main application was not highlighted enough. The main purpose of our work is to implement the torus mapping method for operational data assimilation, specifically for the ACCORD consortium, with the use of OPERA radar dataset. By validating the method against a large independent observation and model reference datasets, we want to demonstrate that the method is robust enough to handle radar observations from a very heterogenous source (OPERA) and provide dealiased data of sufficient quality for use in NWP. To highlight this goal, we propose a slight title change, to "Validation of torus mapping radial wind dealiasing method for use in NWP".

**1. RESPONSES TO GENERAL COMMENTS**

(1)  • Yes, as mentioned above, the main focus of our validation is data assimilation of Doppler winds provided by OPERA programme by EUMETNET. It is a centralized repository for radar data from most European countries with various degrees of preprocessing and quality control provided by individual weather services. This results in a heterogenous set in terms of preprocessing, scanning strategies, radar configurations, etc., that also changes with time. For use in NWP assimilation, we therefore need a very robust method to handle this heterogenous set, without tuning for individual radars. It also should not rely on an external data source. As the intent is operational use, method must also be fast and use little CPU resources. We looked into existing methods and identified the torus mapping method as most promising for this purpose, as it has all the required properties.

- The torus mapping method is very similar (to the first order in $\Delta\theta$) to the V-IVAP method used by Liang et al. 2019, hereafter L19 (see answer 2 and section 4), so for sparse data (in the azimuth direction), the method works well, provided that the azimuth interval used in determining the reference velocities is as big as possible. That is why we used the whole interval (-180,180) in our implementation of the torus mapping method.

- For high shear in vertical direction, the method also works well (see figure 1), because data used in our implementation of the torus mapping is divided into 100 m height intervals, combining data from multiple elevations. For each interval, a separate reference velocity is calculated, allowing for any wind change in the vertical direction.

[Figure]

FIGURE 1. Dealiasing for a high vertical shear case on 4.June 2021 for an elevation with angle $6.3°$ on the Pasja Ravan radar.

- For high shear in the horizontal direction, the results of the torus mapping method are poorer (similar to the V-IVAP method) (see figure 2). It could be augmented by a secondary more local method as in L19, however, since we expect that the remaining incorrectly dealiased data from high shear areas would be filtered out

by the background check as part of the quality control in data assimilation (figure 8 in paper), we do not employ this extra step to reduce computing time.

[Figure]

FIGURE 2. Dealiasing for a high horizontal shear case on 5.May 2021 for an elevation with angle $3.8°$ on the Pasja Ravan radar.

(2)
- As our focus is on a large scale validation and usefulness in assimilation, we did not seek to develop an entirely new method, but rather use an existing one and adapt it for use in assimilation. The reasons for choosing the torus mapping method are explained in (1). The decision to use torus mapping was taken in 2019, when the new V-IVAP/IVAP method by L19 was not published yet, but after a review of the suggested paper, we conclude that the torus mapping method is very similar to the V-IVAP method. In fact, if we expand equations (7) of L19 to the first order in small $\Delta\theta$ and express the azimuth derivative $(V_{r,\theta} - V_{r,\theta-\Delta\theta})/\Delta\theta$ (see section 4), we get the same equation from which the reference wind is determined in both methods (for torus mapping these are equations (5)-(8) in the paper). While we do not apply a second step for local corrections (as the IVAP method in L19), we show that using only torus mapping is enough for assimilation purposes.
- The OPERA programme does not provide any dealiased datasets, apart from radar networks that apply dual- or triple-PRF technique (DE,FR); for those countries,

data is still aliased on the extended Nyquist interval. As far as we know, in AC-CORD no country applies dealiasing operationally. Dealiasing is used at the Swiss meteorological service, but these results are not provided to OPERA and a direct intercomparison is not possible.

(3)  • We agree with the referee that there is a discrepancy between radars introduced with figure 4 in paper and those used for validation as datasets A and B. For a detailed analysis, we wanted to focus on radar networks where aliasing of data is the main problem, to exclude as much as possible the sources of other nonrelated errors (such as noise), which mask the impact of the dealiasing on the results. However, the validation was done on all shown radars and we will provide analyses per radar network (country), further clarifying the selection of datasets for detailed analysis.

• As explained in (2), data from DE and FR are already provided on the extended Nyquist interval. Because we want the dealiasing method to be universal for all OPERA data, the errors that stem from the multi-PRF dealiasing are not specifically treated, but included in the analysis as all other possible errors in the width of statistical distributions.

• The SI dataset is indeed different from dataset A only by filtering for events, but is named differently to make the distinction as SI and DE datasets do not contain colocated pairs.

In figure 8 in paper, we mistakely included only colocated pairs. This will be corrected, as the assimilation procedure works on all data.

(4) Figure 3 in paper is just for illustration of the way our implementation of torus mapping algorithm rejects data, that is why values are not shown to emphasise the rejection, but we will provide explicit dealiasing examples for low Nyquist velocity (see figures 1, 2).

(5) Algorithm specifications and performance was indeed not put in the paper, but it should be and will be included in the paper. Algorithm was written in the Python 3.10 programming language. With it, we performed dealiasing on 10 random samples each containing around 50 3-volume HDF5 files taken from OPERA in the year 2021. The dealiasing was done on a HP EliteDesk 800 desktop computer, with an Intel Core i5-8500 3.00 GHz processor, 16 GB of DD4 RAM and 931 GB HDD. Processing time for one radar volume ranged from 1-15 s, depending on the amount of data contained in the file, on average the processing time for one radar volume was around 3 s.

**2. RESPONSES TO SPECIFIC COMMENTS**

(1) Thanks, will be corrected.

(2) We will include the recommended (more recent) advances and expand this section.

(3) We apply linear wind assumption on height intervals (of 100 m). However, we apply the same assumption to the whole azimuth interval, as we want the method to work also for sparse data as explained in answer to general comment (1). In this, we are similar to the V-IVAP method of L19. A local method is not applied, also explained in answer (1), dealing with high-shear situations.

(4) As explained in answer to general comment (2), the torus mapping method is equivalent (to the first order in $\Delta\theta$) to the V-IVAP method of L19 (which is based on VAD), by using azimuthal variances (azimuthal derivative) of radial velocity to retrieve reference winds.

Compared to L19 V-IVAP implementation, we use the whole azimuth angle interval, with 100 m height levels, 60 m/s threshold on the reference wind. We also do not reject VRAD values below 1 m/s to retain valuable information for data assimilation and do not perform the interpolation from radar coordinates to lat-lon grid and back. We also do not apply the IVAP method step afterwards as explained in answer to general comment (2).

(5) Torus mapping is sensitive to noise and performs poorly if the amount of noise is high as shown with a small example. Because of this we recommended denoising before using the method to dealias data.

The method only has poor performance in horizontal high shear situations such as frontal boundaries and high turbulence situations. As the target application is assimilation, we show that small-scale errors done in these cases will most likely be filtered out by the quality control of data assimilation (also see answer to general comment (1)).

(6) You are correct, we use data from all elevations that fall inside a specific 100 m height interval for determining the reference wind. By using OPERA data, which has data files grouped in 15 min intervals, this means that we use data from multiple consecutive volumes contained in the file (e.g. 3 volumes of 5 minute measurements). This also coincides with our time window used in colocation and the current needs for data assimilation in terms of observation frequency.

(7) As mentioned in answer to general comment (3), we performed the validation for all countries, but chose specific radar networks that have aliasing as their main problem for the detailed analysis. This is done to show the impact of just the torus mapping dealiasing on data without masking the impact with errors from other sources.

As explained in answer to general comment (2), most ACCORD countries do not deliver dealiased data to OPERA, apart from multiple-PRF Nyquist interval extensions.

(8) All radar networks that lie inside our NWP domain were introduced, however, only the ones included in datasets A and B had the aliasing of data as the main source of error (see also answer to general comment (3)). The validation was done on all networks, so we will provide a summary of results for all radar networks to show why the detailed selection was made.

(9) Thanks, the figures will be corrected.

(10) Indeed, the hatching in the panels is not optimal for clarity and will be corrected.

(11) Yes, we only used log scale for dataset B for easier visual interpretation as there, data is less aliased. The use of log scale for dataset A would highlight additional peaks at multiples of Nyquist velocity, so we will use log scales for all figures.

(12) Thanks for pointing out this error. It will be corrected.

(13) We agree with this statement, and will rephrase the sentence and include a summary of analyses for all radar networks that have been excluded, to justify their exclusion.

(14) Acceptance rate as used in the manuscript is the ratio of accepted and all data that enter the data assimilation. It is a function of the dealiasing method applied but also of the assimilation quality control used. So it is difficult to compare this rate by only comparing dealiasing methods.

Dealiasing failure rate is based on an assumption that all aliased data is contained in the side peaks of the difference distributions, against other observations or NWP model. After the dealiasing, the peaks reduce roughly by a factor of 10, which means that 10 percent of initially aliased data remains aliased after applying torus mapping. This can be taken as an estimate of the failure rate.

Compared to failure rates of methods from the recommended literature, this rate is much higher, but it has to be noted that in these methods, failure rates were estimated in a controlled, idealized framework, where references used for truth (S-band radar, model results) were artificially aliased. While this is of course a good methodology for evaluating the theoretical failure rate of a method, we chose the mentioned rough estimate as we do not have a proper truth reference so there are additional factors contributing to the failure rate estimate (NWP errors, measurement errors, natural variability between colocated points). As shown in figure 8 in paper, the failure rate can be compensated by using stricter quality control in assimilation.

(15) Because of these fundamental differences, we compared measurements using statistical distributions of colocated pair differences from large samples. This is often used in data assimilation to estimate quality of new observations without knowing the exact

sources of errors, as the distribution widths for pair differences contain all sources of errors from both measurements in an equal manner. This is also the reason that we used aircraft-sonde pairs as reference in figure 5 in paper, as both are already established measurements used in data assimilation.

The number of folds was examined on a spatial map, as seen in the middle plot of figures 1 and 2.

(16) The denoising was not applied for this analysis, but we demonstrate that it has a big impact on results in case it is centered around zero (e.g. ground clutter). We recommend that in this case, denoising should be done before using the torus mapping method. High-noise levels in data is also one of the reasons why we exclude such datasets from a detailed analysis, as an analysis of such a dataset would not show the effects of the dealiasing, but would be dominated by errors from the noise, to which the torus mapping method is sensitive.

For multi-PRF data with specific noise type, dealiasing is shown to be suitable as can be seen in figure 7 b) in paper, where we have results for DE dataset with dual-PRF.

(17) Thanks, will be corrected.

(18) We chose the torus mapping method, because it is robust, fast and works well for a wide range of situations and radar configurations. It's downside is that it has a poorer performance in horizontal high shear cases, but it nevertheless provides a large increase in number of available wind observations for general data assimilation. As explained above in answer to specific comment (14), the failure rate is estimated differently as in Louf et al. and Feldmann et al and is thus not comparable.

**3. REVISION OF PAPER**

Given the very relevant questions raised by Referee 1, we propose a revision of the paper, where we would:

- Since the purpose of our work is to show that the torus mapping method provides dealiased data of sufficient quality for use in NWP, we will make a slight change in the title and revise the text of the paper to make this purpose clearer.

- We will include more recent references that Referee 1 suggested, with more discussion and compare our method to the similar V-IVAP method.

- We will include individual case studies to show the performance of the algorithm in high shear cases.

- To explain our choice of datasets, we will include analyses from all radar networks and justify our reasons for choosing a subset of data for detailed analysis and revise the text accordingly.

- Improve algorithm implementation description (performance, specifications).

- Correct the figures and other smaller errors as suggested.

**4. COMPARISON OF REFERENCE WIND EQUATIONS**

In L19, their equations (7) are used to determine the components of reference velocity $(\overline{u}, \overline{v})$. In the equations, we can simplify expressions if we assume $\Delta\theta \equiv d\theta \ll 1$:

$$\cos\theta - \cos(\theta - d\theta) \approx -d\theta\sin\theta, \qquad \sin\theta - \sin(\theta - d\theta) \approx d\theta\cos\theta,$$

If further define $dV_{r,\theta} \equiv V_{r,\theta} - V_{r,\theta-d\theta}$, and use the simplified expressions in their equations (7), the first equation becomes:

$$\sum_\Omega dV_{r,\theta}d\theta\cos\theta = \overline{u}\sum_\Omega (d\theta)^2\cos^2\theta\cos\phi - \overline{v}\sum_\Omega (d\theta)^2\sin\theta\cos\theta\cos\phi,$$

and the second equation of (7) is identical to the first. Now we can express the azimuth derivative of radial velocity:

$$\sum_\Omega \frac{dV_{r,\theta}}{d\theta} = \sum_\Omega (\overline{u}\cos\theta - \overline{v}\sin\theta)\cos\phi.$$

To obtain the reference velocities in the torus mapping method, we minimize the square of differences between LHS and RHS of our equation (5) for a chosen subset of data:

$$\sum_\Omega \frac{dF_{3,r,\theta}}{d\theta} = \sum_\Omega (-a\overline{u} + b\overline{v}),$$

where we now use the same notation as in L19 for easier comparison.

The derivative $\partial F_3/\partial\theta$ can be expressed from our equation (4):

$$\frac{dF_{3,r,\theta}}{d\theta} = -\sin(V_{r,\theta}\frac{\pi}{v_{ny}})\frac{dV_{r,\theta}}{d\theta}.$$

If we insert this into our equation (5) and use our equations (6) and (7) for coefficients $a$ and $b$, we get:

$$\sum_\Omega \frac{\partial V_r}{\partial\theta} = \sum_\Omega (\overline{u}\cos\theta - \overline{v}\sin\theta)\cos\phi.$$

So theoretically, both methods use the same relation (to the first order in $\Delta\theta$) to determine the reference velocity. Of course, the differences are in implementation; L19 solves a system of two equations, while we use a least squares minimization approach. Second difference is in the numerical method of derivative calculation, above we see that in L19, the method

would be left differences, while we use central differences. The third difference is, that the torus mapping method does not numerically calculate the derivative $dV_{r,\theta}/d\theta$ directly, but calculates $dF_{3,r,\theta}/d\theta$, which contains an extra factor of $\sin(V_{r,\theta}\frac{\pi}{v_{ny}})$.

---

## Author Comment (AC2)

**Response to Alain Protat: Validation of torus mapping method for dealiasing Doppler weather radar velocities**

The authors would like to thank dr. Protat for constructive comments.

As in the answer to Referee 1, the main application, which is usage of dealiased data in data assimilation for NWP, was not highlighted enough. In light of this usage, some issues raised by Referee 2 (large error rates, applicability in convective cases) are of a lesser concern than if the goal was validation of a new method.

The error estimation provided in the paper is not comparable to the estimation cited by the Referee, as the methods of estimation are different and are done on very different datasets. In the absence of relevant truth data, a colocation technique used in the study allows for a relative intercomparisons of observation quality using PDFs, which is often used to validate new datasets in data assimilation. As datasets included in our validation are provided by OPERA and are not preprocessed by us, they can still contain unwanted characteristics, such as a large amount of noise, ground clutter, etc., all of which contribute to error estimation in the paper. On the other hand, error estimates quoted in publications mentioned by the referee evaluate the error rate of the method in idealized conditions, often on synthetically aliased datasets, where measurement errors are neglected, have short validation periods and preprocessed or reduced datasets (denoising, exclusion of data under 1 $m/s$, etc.),

In NWP, convection is only partly resolved, so the details of convective situations are not fully extracted from radar measurements by the assimilation scheme, because winds can only be assimilated at scales that the model can resolve. Given the current resolutions of operational limited area NWP models, the goal is to describe mesoscale and larger convective weather systems. In most of these situations, a method using a linear wind assumption in dealiasing is satisfactory.

**1. RESPONSES TO GENERAL COMMENTS**

(1) We agree that methods that produce identical equations to the ones in the torus mapping method are derived in papers cited by the referee (see section 4 in response to RC1). Our method of choice was the torus mapping by Haase et al, 2004, which was cited, but we agree that citations of these papers should be included.

(2) We agree and will add descriptions of cases where the torus mapping method does not work well and simulations showing inadequacy of the method for nonlinear winds. As explained in the response to general comment (1) of Referee 1, an extra step should be implemented to correct these situations, but we decided not to employ this extra step to reduce computing time.

(3) We will include the suggested references in the paper.

(4) Figure (3) in the paper shows exclusion of events that happen because of three reasons. First is the exclusion because of the centre-difference scheme, which excludes points without two neigbours in the azimuth direction. As the case shown in figure (3) contains a large amount of noise which consists of points without neighbours, which are rejected. Second and third reasons for rejection are noncovergence of the wind model minimization or too few points in the height interval. Both reject points from the whole interval, which is seen in regions further away from the radar site. While the exclusion percentage is high, this is not a problem for assimilation purposes, as radar measurements present a very high number of dense observations, which need to be thinned considerably in order to satisfy computational limitations when assimilating them into the model.

(5) While we agree that aircraft avoid areas with severe convection, we note that aircraft measurements used in the validation are from vicinity of precipitation areas, where radial wind measurements are avalable. These areas likely include stratiform and convective cases.

(6) While we combined radars into datasets based on similar values of Nyquist velocities to study the relevance of dealiasing, we also performed the validation on radars for

each country/provider, which typically have the same Nyquist velocities. Nevertheless, wrongly dealiased data for each Nyquist velocity produce a distinct peak in the PDF, as shown in e.g. figure (7).

As explained in the answer to specific comment (14) and (15) to RC1, we cannot use the error rate as a metric to estimate the accuracy of the technique as we do not have a proper truth reference. For the torus mapping technique Haase et al. cite error rates of the method as 0.2% for stratiform and 4.2% for convective cases, counting just cases where dealiasing was needed.

(7) Our goal is to apply the dealiasing algorithm on a large heterogenous dataset provided by OPERA for use in data assimilation. We do not perform any individual preprocessing, as preprocessing is provided operationally by OPERA. For this reason, the datasets included in our validation can still contain unwanted characteristics. Furthermore, the 10% error rate is expected to decrease significantly during the quality control step before assimilation as shown in figure (8). As explained in the previous answer, the error rate as used in other analyses is not applicable in our case, as we do not have the proper truth reference values. Our error rate estimation covers all mentioned effects, collected on data in 1 year.

(8) We would like to emphasize that in this comparison, NWP is not used as a truth, but as a reference for a relative intercomparison with other data types. We agree that the convection is not fully resolved in this intercomparison, as it is not resolved by aircraft and radiosonde data. On the other hand, all these datasets resolve mesoscale circulations. We will add this discussion to the paper.

**2. REVISION OF PAPER**

Given the very relevant questions raised by Referee 1 and Referee 2, we propose a revision of the paper, where we would:

- Since the purpose of our work is to show that the torus mapping method provides dealiased data of sufficient quality for use in NWP, we will make a slight change in the title and revise the text of the paper to make this purpose clearer.
- Will add descriptions of cases where the torus mapping method does not work well and simulations showing inadequacy of the method for nonlinear winds.
- We will include more recent references that Referee 1 and Referee 2 suggested, with more discussion and compare our method to the similar V-IVAP method.
- We will include individual case studies to show the performance of the algorithm in high shear cases.
- To explain our choice of datasets, we will include analyses from all radar networks and justify our reasons for choosing a subset of data for detailed analysis and revise the text accordingly.
- Improve algorithm implementation description (performance, specifications).
- Correct the figures and other smaller errors as suggested.